# `PokéChamp`: an Expert-level Minimax Language Agent

**Seth Karten** [* 1]   **Andy Luu Nguyen** [1]   **Chi Jin** [1]

## Abstract

We introduce `PokéChamp`, a minimax agent powered by Large Language Models (LLMs) for Pokémon battles. Built on a general framework for two-player competitive games, `PokéChamp` leverages the generalist capabilities of LLMs to enhance minimax tree search. Specifically, LLMs replace three key modules: (1) player action sampling, (2) opponent modeling, and (3) value function estimation, enabling the agent to effectively utilize gameplay history and human knowledge to reduce the search space and address partial observability. Notably, our framework requires no additional LLM training. We evaluate `PokéChamp` in the popular Gen 9 OU format. When powered by GPT-4o, it achieves a win rate of 76% against the best existing LLM-based bot and 84% against the strongest rule-based bot, demonstrating its superior performance. Even with an open-source 8-billion-parameter Llama 3.1 model, `PokéChamp` consistently outperforms the previous best LLM-based bot, Pokéllmon powered by GPT-4o, with a 64% win rate. `PokéChamp` attains a projected Elo of 1300-1500 on the Pokémon Showdown online ladder, placing it among the top 30%-10% of human players. In addition, this work compiles the largest real-player Pokémon battle dataset, featuring over 3 million games, including more than 500k high-Elo matches. Based on this dataset, we establish a series of battle benchmarks and puzzles to evaluate specific battling skills. We further provide key updates to the local game engine. This work establishes Pokémon as a benchmark to integrate LLM technologies with game-theoretic algorithms addressing general multi-agent problems. Videos, [code], and [dataset] are available [online].

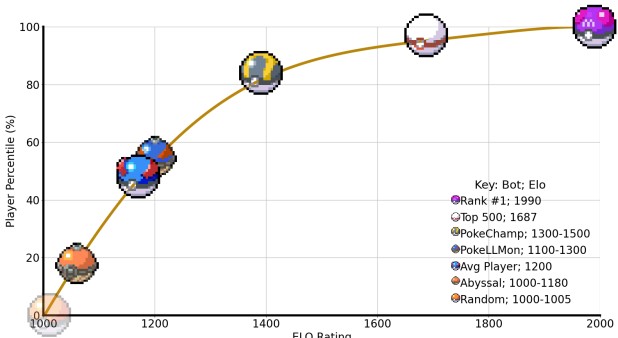

*Figure 1.* `PokéChamp` achieves the 70%-90% percentile of players and a 1300-1500 Elo rating against real players. Higher Elo and percentile denote better performance.

## 1. Introduction

Realizing superintelligence or "takeoff" demands a cycle of recursive self-improvement in which each model iteration can outplay the last—a capacity currently bottlenecked by the strategic search and coordination challenges we observe in competitive games. Prior work in reinforcement learning has achieved significant success, attaining superhuman performance in a wide range of games, including Chess, Go, and Poker, through extensive imitation learning and self-play (Campbell et al., 2002; Silver et al., 2017; 2016; Brown & Sandholm, 2019; 2018; Vinyals et al., 2019; Berner et al., 2019). However, these approaches typically require substantial task-specific training and engineering. In contrast, Large Language Models (LLMs), which often function as generalist agents, have demonstrated remarkable capabilities across various domains. Language agents can leverage prior knowledge of game strategies and generalize to new situations without additional training.

Despite their promise, recent studies highlight limitations in the planning capabilities of text-based language agents (Topsakal & Harper, 2024). These agents frequently underperform compared to rule-based heuristic bots in game environments (Küttler et al., 2020) and struggle to grasp basic game mechanics (Hu et al., 2024b). To overcome these challenges and further investigate the potential of LLMs in complex game environments, this paper focuses on a fast-paced, strategy-rich, and challenging two-player competitive game—Pokémon battles.

---

[*]Equal contribution  [1]Princeton University. Correspondence to: Seth Karten <sethkarten@princeton.edu>.

*Proceedings of the $42^{nd}$ International Conference on Machine Learning*, Vancouver, Canada. PMLR 267, 2025. Copyright 2025 by the author(s).

Pokémon battles present a unique and formidable challenge for AI systems. With over 1000 Pokémon species, each possessing unique abilities, moves, typing, and stats, the state space complexity is estimated to be on the order of $10^{354}$ for the first turn alone (The-Third-Build, 2022). The game features partial observability, where information about the opponent's team is gradually revealed through gameplay, maintains this vast search space throughout the battle. Furthermore, Pokémon battles can last anywhere from 6 to over 100 turns, making exhaustive tree search over all possible branches computationally intractable.

We argue that an informative prior can help constrain minimax search to the space of human strategies. LLMs, trained on diverse datasets that include Pokémon-related information, offer a promising foundation for this approach. To harness the potential of LLMs effectively, we seek to develop an agent capable of:

1. Proposing strategic actions to provide diverse and human-like strategies.

2. Accurately modeling the opponent based on their move history, team composition, and skill level.

3. Evaluating and reflecting internally on planned game trajectories.

To achieve these objectives, we introduce PokéChamp, a competitive language agent that achieves human-expert level performance in two-player turn-based Pokémon battles. PokéChamp leverages a large language model to power minimax tree search algorithm by replacing three of its key modules with LLMs: (1) player action sampling, (2) opponent modeling, and (3) value function estimation. This enables our agent to effectively utilize gameplay history and human knowledge to reduce the search space and tackle partial observability. Within our framework, the LLM functions as a black box, allowing for flexibility in selecting any frontier model based on budget and computational resources. Our approach does not require additional training or fine-tuning of LLM on Pokémon-specific data, relying instead on the LLM's pre-existing knowledge and our novel integration with game-theoretic planning algorithms.

To enable effective planning, we developed a world model that approximates game transition and addresses the challenges of partial observability and intricate game mechanics. This world model incorporates a one-step lookahead that mathematically computes core game dynamics and leverages historical data from real player games to estimate likely stats for the opponent's team. Our Pokémon battling dataset, containing over 3 million games across various skill levels and game modes, provides a rich source of information for opponent modeling and strategy development.

We evaluate PokéChamp through a comprehensive set of experiments designed to assess its performance across various competitive scenarios. Our evaluation includes arena-style comparisons against established Pokémon bots, encompassing both heuristic-based approaches and the state-of-the-art LLM-based agent PokéLLMon (Hu et al., 2024b). To gauge PokéChamp's versatility, we conduct performance analyses in two popular game modes: Generation 8 Random Battles and Generation 9 OverUsed Meta. These formats present distinct challenges, with Random Battles testing adaptability to unpredictable team compositions and OverUsed (OU) format battles examining strategic depth with carefully crafted teams. Finally, to assess real-world applicability, we pit PokéChamp against human players in online ladder battles, providing insights into its performance against skilled opponents in a dynamic, competitive environment.

Our results demonstrate that PokéChamp significantly outperforms existing bots and AI agents, achieving a 76% win-rate against the strongest LLM-based bot and an 84% win-rate against the most advanced heuristic bot in the Generation 9 OverUsed Meta. Notably, PokéChamp using the open-source 8 billion parameter Llama 3.1 model consistently wins (64%) against the prior strongest LLM-based bot utilizing GPT-4o, highlighting the effectiveness of our approach even with smaller language models. In online ladder battles, PokéChamp attains an expert-level projected Elo rating of 1300-1500, placing it within the top 30%-10% of competitive players. This performance demonstrates the agent's ability to compete at a high level against skilled human opponents in a dynamic, partially observable environment.

In addition to our main contributions, we present several supplementary advancements that enhance the scope and impact of our research. We introduce the largest Pokémon battling dataset to date, encompassing over 3 million games, with more than 500,000 high Elo matches, providing an unprecedented resource for analyzing competitive play patterns and strategies. To rigorously evaluate battling proficiency, we develop a comprehensive series of benchmarks derived from real player data and meticulously crafted puzzles, designed to assess specific battling abilities and decision-making skills. Furthermore, we implement crucial updates to the local game engine, significantly improving its accuracy and performance, thereby ensuring a more faithful representation of official battle mechanics and enabling more reliable simulations and evaluations.

Through PokéChamp, we demonstrate the potential of integrating LLMs with game-theoretic planning algorithms to achieve expert-level performance in complex, partially observable environments without task-specific training. Our work opens new avenues for research in competitive multi-agent settings and pushes the boundaries of AI performance in strategic game play.

## 2. Competitive Pokémon

Competitive Pokémon battling presents a complex, partially observable Markov game environment with a vast state space. Two players strategically deploy teams of six Pokémon in turn-based combat, with only one Pokémon active per player at a time. Each turn, a player can choose to attack, reducing the opponent's Pokémon's health points, or switch to another Pokémon in their team. The game concludes when all Pokémon on one side have their health points reduced to zero. The state space is immense, with an estimated $10^{354}$ possibilities for the first turn alone, stemming from over 1000 Pokémon species with varied types, stats, and movesets. Players face asymmetric observation spaces, having complete information about their own team but only partial information about their opponent's. Strategic decisions must therefore be made under uncertainty.

Team construction, or "teambuilding" (as illustrated in Figure 3), is a critical pre-battle strategic element. For each Pokémon, players configure moves, abilities, held items, a nature, and a specific distribution of Effort Values (EVs) and Individual Values (IVs) that determine the Pokémon's statistics. The observation and action space are depicted in Figure 2. The action space consists of move selection, Pokémon switching, and generation-specific mechanics like Terastallization. The transition function incorporates stochastic elements like move accuracy and damage calculation, while rewards are binary (win or loss). The competitive metagame evolves as players discover new strategies, making adaptation essential. This complexity, combined with partial observability, makes competitive Pokémon an ideal testbed for game-theoretic research.

### 2.1. Pokémon Showdown Platform

Pokémon Showdown is a widely-used online battle simulator that implements the official battle mechanics, providing a platform for various competitive formats. It allows players to engage in battles without the need for in-game breeding or training, facilitating rapid experimentation and competitive play. Showdown supports multiple battle formats, including Singles, Doubles, and various tiered formats that group Pokémon based on their usage and perceived strength. The platform's accessibility and faithful recreation of game mechanics have made it a cornerstone of the competitive Pokémon community and a valuable tool for AI research in complex game environments.

Generation 9 OverUsed (OU), the latest generation's most popular competitive format, exemplifies the intricate balance between diversity and strategic depth in Pokémon battling. The OU format bans exceptionally powerful Pokémon while maintaining a wide pool of viable options, creating a rich and evolving metagame. This format introduces mechanics like Terastallization, which allows a Pokémon to change its

type once per battle, significantly altering type matchups and strategic considerations. Furthermore, the sheer number of possible Tera types (all 18 Pokémon types) for each Pokémon drastically increases the state and action spaces. As a result, agents need to navigate an evolving landscape of popular Pokémon and counter-strategies, balancing individual strengths with synergistic team compositions while accounting for the dynamic type changes introduced by Terastallization.

Each player has a total clock time of 150 seconds for the entire match. In addition, each turn has an incremental time limit of 15 seconds. If a player exceeds either the total clock time or the incremental turn time, they automatically lose the match. This time constraint adds another layer of complexity, requiring efficient decision-making and preventing exhaustive search strategies.

## 3. Mathematical Formalization

We can formalize Pokémon battles as a partially observable Markov game (POMG) defined by the tuple $(\mathcal{S}, \mathcal{X}, \mathcal{Y}, \mathcal{A}, \mathcal{B}, H, P, r)$, where $\mathcal{S}$ is the latent state space. $\mathcal{X}$ and $\mathcal{Y}$ are the observation spaces (infosets) for the max-player and the min-player, respectively. The observation contains only information accessible to each player. $\mathcal{A}$ and $\mathcal{B}$ are the action spaces for the max-player and min-player. $H$ is the horizon length. $P : \mathcal{S} \times \mathcal{A} \times \mathcal{B} \to \Delta(\mathcal{S})$ is the transition function, where $P(s'|s, a, b)$ denotes the probability of transitioning to state $s'$ if starting from state $s$, and the max-player and the min-player take action $a, b$ respectively. $r : \mathcal{S} \times \mathcal{A} \times \mathcal{B} \to [0, 1]$ is the reward function. We say a POMG has a *tree structure* and *perfect recall* if the followings hold. (1) Tree structure: for any state $s_h$ at step $h$, there exists a unique history $(s_1, a_1, b_1, ..., s_{h-1}, a_{h-1}, b_{h-1})$ of past states and actions that leads to $s_h$. (2) Perfect recall: any infoset $x_h$ of the max-player at step $h$, there exists a unique history $(x_1, a_1, b_1, ..., x_{h-1}, a_{h-1}, b_{h-1})$ of past states and actions that leads to $x_h$. The same condition applies symmetrically to the min-player. Throughout this paper, we assume the learner be the max-player who maximize the reward, and let the opponent be the min-player.

In Pokemon battle, a state contains the history of the current game and the complete information of both players' Pokémon teams (including Pokemon health, stats, items, status ailments, and other attributes), while an observation contains the history of the current game and the complete information of agent's team but only the partial information of the opponent's team. That is, an observation can correspond to multiple underlying states. The Pokemon battle is a tree-structured POMG with perfect recall.

**Policy and Nash equilibrium**  A policy for the max-player is denoted by $\mu : \mathcal{X} \to \Delta(\mathcal{A})$, which is a map from ob-

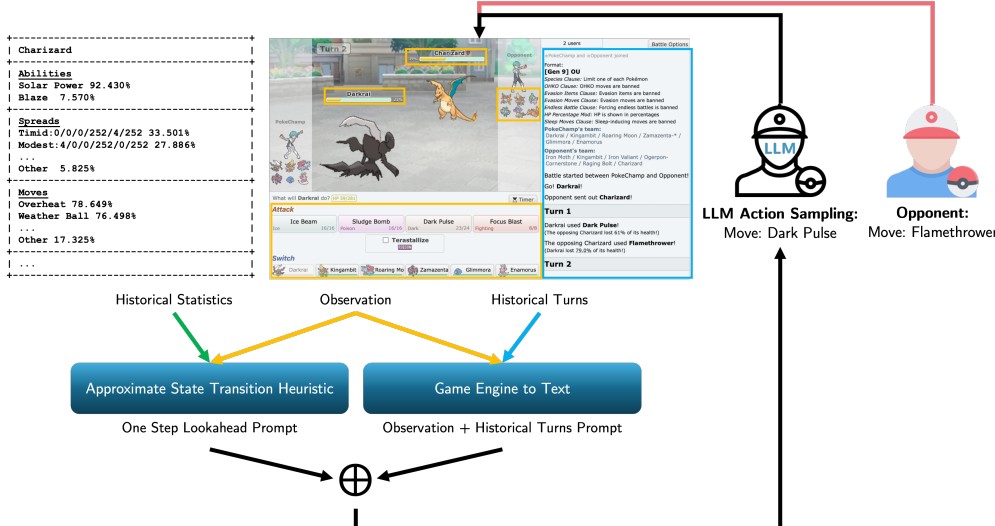

*Figure 2.* `PokéChamp` uses one-step lookahead prompts to gain admissible heuristic information regarding the likely effect of actions under the current metagame.

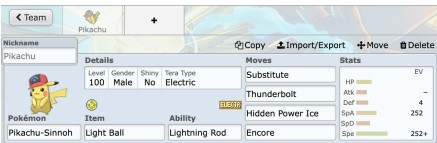

*Figure 3.* An example of teambuilding for competitive Pokémon. A player must decide on six Pokémon for their team. For each Pokémon, a player must configure the item, ability, moves, stats (EVs/IVs), and nature.

servation space to the space of distribution over actions. Similarly, a policy for the min-player can be denoted as $\nu : \mathcal{Y} \to \Delta(\mathcal{B})$. Denote the game trajectory over $H$ steps to be $(s_1, a_1, b_1, \ldots, s_H, a_H, b_H)$, then the value function for a policy pair $(\mu, \nu)$ is defined as:

$$V^{\mu,\nu} = \mathbb{E}_{\mu,\nu}\left[\sum_{h=1}^{H} r(s_h, a_h, b_h)\right]$$

where the expectation is taken over the trajectories following policies $\mu, \nu$.

The Nash equilibrium is defined as a pair of policies $(\mu^*, \nu^*)$ that no player has incentive to change her strategy while the other players keep theirs unchanged. That is,

$$\inf_{\nu} V^{\mu^*,\nu}(s) = V^{\mu^*,\nu^*}(s) = \sup_{\mu} V^{\mu,\nu^*}(s)$$

The minimax theorem holds in our setting, and Nash equilibrium achieves the minimax value:

$$\sup_{\mu}\inf_{\nu} V^{\mu,\nu}(s) = V^{\mu^*,\nu^*}(s) = \inf_{\nu}\sup_{\mu} V^{\mu,\nu}(s) \quad (1)$$

Combining two equations, it is not hard to see that Nash equilibrium is also the optimal policy against the adversarial opponent. This formalization captures the essential elements

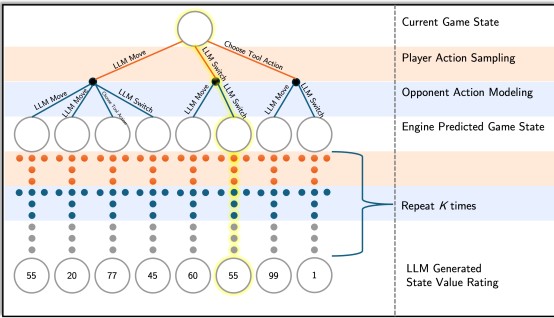

*Figure 4.* `PokéChamp` replaces three components of minimax tree search with LLM-based generations: (1) sampling potential actions for the player corresponding to the first part of the edge between states., (2) modeling the opponent and sampling opponent actions corresponding to the second part of the edge between states, and (3) generating a potential game state value based on the depth $K$ cutoff. `PokéChamp` provides the action with the best minimax value to be used in battle.

of Pokémon battles, including partial observability, turn-based actions, and the goal of maximizing expected rewards, while providing the necessary context for the methodology section.

## 4. Agent Architecture and Application

`PokéChamp` implements a novel approach that leverage the power of large language models and integrate them into minimax tree search. We first describe the generic framework which applies to general two-player zero-sum games. We then discuss specializations we make to adapt our framework for Pokémon battle.

**Minimax tree search framework.** Due to computation constraints and tight inference time requirement, we focus on pure strategies and compute the best action that is safe

even against the adversarial play of the opponent. The overall strategy corresponds to minimax tree search which is described at Figure 4:

At step $h$, the learner observes an infoset $x_h$, chooses an action $a_h$, and the opponent chooses an action $b_h$. The game transition to the new infoset $x_{h+1}$. We repeat this process until game terminates at step $H$, which leads to a tree of depth $H - h$ where each path of the tree corresponds to a $3(H - h)$-tuple $(a_h, b_h, x_{h+1}, \ldots, a_{H-1}, b_{H-1}, x_H)$. In the end, we receive the reward $r(x_H)$ for every infoset $x_H$ at the step $H$. The minimax tree search algorithm recommend action $\widehat{a}_h$ to the learner at $x_h$ according to the following criteria:

$$\widehat{a}_h = \arg\max_{a_h} \min_{b_h} \mathbb{E}_{x_{h+1}} \ldots \max_{a_{H-1}} \min_{b_{H-1}} \mathbb{E}_{x_H} r(x_H)$$

It is not hard to see that, the size of the tree grows exponentially with respect to the search depth $H - h$, and thus a comprehensive search over the entire tree quickly becomes infeasible.

We propose a novel minimax agent framework that integrates LLMs into three key modules of classical minimax tree search, significantly enhancing its performance:

- *Player Action Sampling*: We provide the LLM with key information about the current infoset and prompt it to sample a small set of viable actions for tree expansion. This aggressively prunes the search tree, reducing computational costs.
- *Opponent Modeling*: Similarly, we prompt the LLM to sample the most likely opponent actions. Unlike player action sampling, opponent modeling is more complex due to partial observability, requiring the LLM to infer hidden states based on the player's observations.
- *Value Function Estimation*: Instead of expanding the search tree to the end of the game, we limit expansion to a depth of $k$ steps and use LLMs as value function to evaluate infosets at this time step, replacing terminal rewards. It is challenging to estimate this value accurately through traditional methods, as it requires a deep understanding of game mechanics and effective state embeddings. In contrast, LLMs leverage common sense and gameplay knowledge from the internet, serving as an effective approximation for value functions.

### 4.1. PokéChamp

We now describe the implementation of PokéChamp and the design of its three key modules.

**Approximate game transition** One challenge in our system is simulating game transitions, represented as $P_h(s_{h+1}|s_h, a_h, b_h)$, under partial observability. In our setting, the learner only has access to the observation $x_h$ rather than the latent state $s_h$. To infer hidden information and approximate latent state, we leverage the statistical data from Pokémon Showdown, including Pokémon move pools, EV spreads, item usage, etc., aggregated from gameplay over a given period. Additionally, we incorporate LLM predictions to estimate hidden opponent variables, particularly the attack (A) and defense (D) stats, inferred from game history. The relationship between these stats and damage output is captured by the following equation:

$$\text{Damage} = \left( \frac{1}{50} \left( \left( \frac{2}{5} \cdot \text{Level} + 2 \right) \cdot \text{Power} \cdot \frac{A}{D} \right) + 2 \right) \cdot M$$

where $A$ represents the attacker's relevant attack stat and $D$ the defender's relevant defense stat. The $M$ term encapsulates various game-specific modifiers (detailed in the appendix, equation 2). After predicting $s_h$, we obtain $s_{h+1}$ by simulating the game using our local Showdown simulator. Notably, the actual Pokemon game features stochastic transition—for example, many attack moves have a probability of missing. To alleviate computational burden, we adopt a simplified approach that computes the expected value within these transitions.

**Player action sampling** For a given infoset $x_h$ at step $h$, PokéChamp generates a set of candidate actions $\{a_h^i\}_{i=1}^m \subset \mathcal{A}$ to form the edges of the minimax search tree. The input prompt for action sampling includes:

- **Team strategy:** An LLM-generated overall strategy based on both players' teams.
- **Observable state:** $x_h \in \mathcal{X}$, including the player's team, items, and visible opponent Pokémon.
- **Battle history:** Information from the last $N$ turns, representing the perfect recall assumption.
- **Approximate State Transition Heuristic:** We use the approximate state transition to create an admissible heuristic for each player action. Using a one step lookahead from the approximate state transition, we can calculate the least number of turns to knock out (KO, a term for defeating the opponent's current Pokémon). This information is commonly available and used by players on the online ladder on Pokémon Showdown.
- **Available actions:** The set of legal actions $\mathcal{A}_h(x_h) \subset \mathcal{A}$ under the current info set.

In addition to LLM generated actions, we also include a few candidate actions from our tools, including (1) the top move choice from our one step lookahead and (2) the top switch choice from the Abyssal bot.

**Opponent modeling** To address the partial observability of the opponent's actions and hidden state information, we employ a combination of historical data analysis and LLM-based prediction:

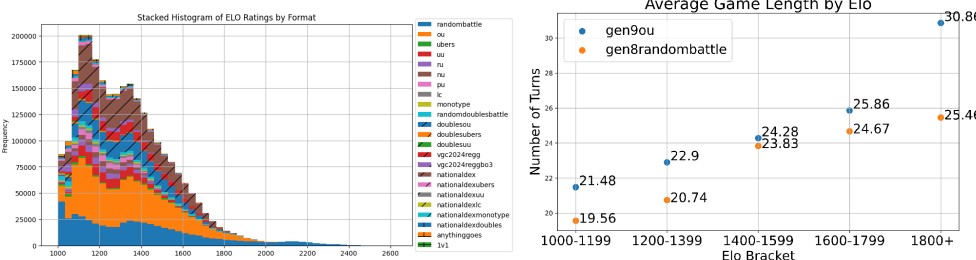

*Figure 5.* **Left:** Elo distribution for collected battles across game formats. **Right:** Relationship between game length and player Elo rating.

- **Stat estimation:** For unknown opponent stats ($A$ and $D$ in equation 4.1), we use historical data to estimate the likelihood of different stat distributions. This allows us to approximate the true state $s_h \in \mathcal{S}$ given the observation $y_h \in \mathcal{Y}$.
- **Action prediction:** The LLM generates likely opponent actions $b_h \in \mathcal{B}$ based on a prompt similar to the action sampling process, but focused on the opponent's perspective.

**Value function** Due to the computational constraints of live gameplay (150 seconds per player, with up to 15 additional seconds per turn), we employ an LLM-generated value function to evaluate leaf nodes at depth $k$ of our minimax tree. The LLM generates a score based on the following criteria:

- **Positive factors:** Effectiveness of current moves, number of remaining Pokémon, and win probability.
- **Negative factors:** Excessive switching, opponent move effectiveness, speed disadvantage, opponent's remaining Pokémon, and strength of remaining opponent Pokémon.

We then compute the minimax optimal action $\widehat{a}_h$ according to:

$$\widehat{a}_h = \arg\max_{a_h} \min_{b_h} \mathbb{E}_{x_{h+1}} \cdots \max_{a_{h+k-1}} \min_{b_{h+k-1}} \mathbb{E}_{x_{h+k}} V(x_{h+k})$$

By combining these three components - action sampling, opponent modeling, and value function approximation - PokéChamp effectively navigates the complex, partially observable state space of Pokémon battles, approximating optimal play within the constraints of real-time gameplay.

## 5. Evaluation

In this section, we describe our evaluation methods which evaluate PokéChamp via both offline dataset and online games on Showdown platform.

**Pokémon Battling Dataset.** We compiled a comprehensive dataset of over 3 million Pokémon battles from the Pokémon Showdown platform. This dataset serves as a rich source of information for estimating transition probabilities

$P_h(s_{h+1}|s_h, a_h, b_h)$ and opponent policies $\nu_h(\cdot|y_h)$. The dataset includes over 3 million game replays across various formats, with more than 500,000 high-Elo games (Elo > 1600). It contains detailed information on team compositions, move choices, and battle outcomes, providing a robust foundation for our analysis and modeling efforts.

Figure 5 illustrates the distribution of Elo ratings and game lengths across different formats in our dataset. The left panel shows a stacked histogram of Elo ratings for various game formats, revealing a multi-modal distribution with two primary modes at approximately 1150 and 1350 Elo. This distribution provides insights into the skill levels of players across different game modes. The right panel presents a scatter plot of average game length versus Elo rating, demonstrating a general trend where higher Elo matches typically have longer durations. This relationship suggests that more skilled players engage in more complex and protracted battles.

### 5.1. Action Prediction

To evaluate the effectiveness of our modules of player action sampling and opponent modeling, we conducted experiments on predicting human actions using our collected dataset. This task is particularly challenging due to the partial observability of the game state.

The replay data is collected from a spectator's perspective, where key information such as EV spreads and item choices for **both** players remains hidden. This contrasts with the standard player perspective, where only the opponent's team information is obscured, while the player's own team details are fully accessible. To bridge this gap, we reverse-engineered the moves, team compositions, and stats from the replay data. We then supplemented this information with additional switching and move options based on historical likelihood. We use this processed data as proxy to real gameplay with a standard player perspective. This processed data was then fed into our prompt generator, enabling our bot to predict both player and opponent actions.

We compared the accuracy of these predictions with the historical data across various Elo ratings. Table 1 presents the results of our action prediction experiments. The player

*Table 1.* Player and opponent action prediction accuracy by Elo rating. Random baseline performance is 7% for player prediction and less than 1% for opponent prediction due to partial observability.

|  | **Elo** | Top 1 | Top 2 | Top 3 | Top 4 | Top 5 |
|---|---|---|---|---|---|---|
| Player's action prediction | 1200 | 30% | 40% | 48% | 53% | 58% |
|  | 1400 | 26% | 23% | 30% | 32% | 43% |
|  | 1600 | 27% | 30% | 39% | 44% | 53% |
|  | 1800 | 30% | 42% | 55% | 62% | 66% |
| Opponent's action prediction | 1200 | 16% | 30% | 40% | 46% | 53% |
|  | 1400 | 16% | 17% | 20% | 26% | 39% |
|  | 1600 | 13% | 21% | 26% | 35% | 40% |
|  | 1800 | 15% | 29% | 40% | 50% | 53% |

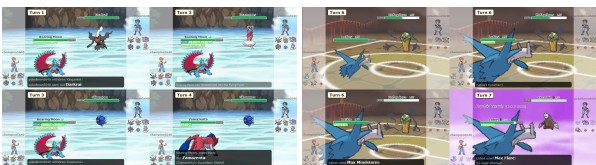

*Figure 6.* **Left:** `PokéChamp` utilizes Terastallization to change type matchups. **Right:** `PokéChamp` employs Dynamax to increase hit points and move power, enabling consecutive knockouts.

prediction accuracy for `PokéChamp` varies between 26% and 30% as Elo increases, while the opponent prediction accuracy is lower, ranging from 13% to 16%. These results demonstrate that predicting opponent actions is more challenging given the limited state information available.

We can see that LLM prediction is significantly better than random prediction (7% for player, 1% for opponent) in both cases. It's worth noting that there may be multiple "correct" actions in many situations, as several strategies could be equally viable. Therefore we included in the table the performance for top $k$ prediction, and observe the accuracy increases significantly as $k$ increases.

### 5.2. Evaluating on Small Game Puzzles

To assess `PokéChamp`'s ability to navigate complex game states and make optimal decisions, we developed a series of game puzzles as benchmarks. These puzzles are designed to test the agent's understanding of core game mechanics and its ability to approximate optimal policies $\mu^*$ and $\nu^*$.

**1v1 Battles.** We created a benchmark of 1,000 1v1 battle scenarios to assess `PokéChamp`'s ability to find optimal move sequences. Each scenario $s_1 \in \mathcal{S}_1$ is carefully selected to ensure a feasible win condition, allowing us to evaluate the agent's performance in finding the optimal policy $\mu^*$. We selected matchups from the gen8randombattles meta, with each matchup consisting of one Pokémon on each team. To ensure a feasible win condition, we rejected samples that could not be won by the Abyssal bot. However, due to the stochastic nature of move damage, this does not

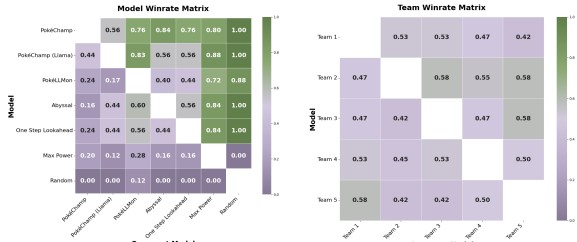

*Figure 7.* **Left:** Pairwise win rates in Gen 9 OU. **Right:** Impact of team composition on win rates.

guarantee a 100% win rate in every instance.

The results of this benchmark show `PokéChamp`'s performance of 86% winrate is higher than PokéLLMon's of 76%. Our method achieves a 10% higher win rate. Since this is a constrained setting that does not require Pokémon switching, a key factor contributing to our agent's superior performance is its effective utilization of one-step lookahead.

**Puzzles for special mechanics: Terastallization and Dynamax.** To evaluate `PokéChamp`'s ability to utilize generation-specific game mechanics, we developed puzzles focusing on Terastallization and Dynamax. These mechanics introduce additional complexity to the state space $\mathcal{S}$ and action space $\mathcal{A}$. Our world model and prompting mechanism for `PokéChamp` allow it to understand and use these generation-specific game mechanics effectively. The agent is informed about the mechanics' effects, and the one step lookahead provides information about different outcomes when using these mechanics.

Figure 6 illustrates `PokéChamp`'s proficiency in leveraging these mechanics to gain a strategic advantage. In the left panel, `PokéChamp` demonstrates its understanding of the changed weakness of Roaring Moon after Terastallization, deciding to switch after Glimmora is chosen. In the right panel, `PokéChamp` employs Dynamax to increase its hit points and move power, enabling it to knock out two Pokémon in succession.

These benchmarks demonstrate `PokéChamp`'s ability to

*Table 2.* Performance in Gen 9 OU battles with terastallize mechanic and custom teams.

| Method | LLM | Win Rate vs. Abyssal (%) | Elo | Avg. # Turns |
|---|---|---|---|---|
| **PokéChamp** | GPT-4o | **84** | **1268** | **15.7** |
| PokéChamp | Llama 3.1:8b | 56 | 1204 | 16.9 |
| PokéLLMon | GPT-4o | 40 | 1020 | 22.6 |
| Abyssal | N/A | N/A | 1117 | 17.9 |
| One Step Lookahead | N/A | 44 | 1107 | 17.9 |
| Max Power | N/A | 16 | 885 | 19.5 |
| Random | N/A | 0 | 399 | 21.2 |

*Table 3.* Gen 9 OU mirror matchups without terastallize mechanic.

| Method | LLM | Win Rate vs. Abyssal (%) |
|---|---|---|
| **PokéChamp** | **GPT-4o** | **90** |
| PokéChamp | Llama 3.1:8b | 83 |
| PokéLLMon | GPT-4o | 60 |
| One Step Lookahead | N/A | 56 |

navigate complex game states and make strategic decisions that approximate optimal policies in the presence of special mechanics. The agent's performance in these puzzles highlights its capacity to adapt to various game scenarios and utilize advanced mechanics to gain a competitive edge.

### 5.3. Evaluating on Full Games

In this section, we evaluate `PokéChamp`'s performance in the Gen 9 OU (OverUsed) format, which includes the terastallize mechanic and uses custom teams. We evaluate `PokéChamp` against other baselines as well as on the online ladder against real human players. Additional experiments for the Gen 8 Random Battles format are provided in Appendix D.

**Against other bots and agents.** We compare `PokéChamp` against state-of-the-art open-source bots and LLM-based agents. Each experiment consists of at least 25 matches between any two methods, resulting in a minimum of 100 games per method for Elo calculations. The LLM agents utilize either Llama3.1:8b (Dubey et al., 2024) or GPT-4o-2024-05-13 (Achiam et al., 2023). Our baselines include:

- **PokéLLMon** (Hu et al., 2024b): An LLM-based agent using self-consistency prompting.
- **Abyssal Bot**: A rule-based heuristic bot used in official Pokémon games.
- **One Step Lookahead Bot**: Our admissible heuristic bot using the approximate state transition from Section 4.1.
- **Max Power Bot**: A bot that selects moves based solely on power level.

- **Random**: A bot that selects actions randomly.

`PokéChamp` with GPT-4o achieves the highest performance, with an 84% win rate against the Abyssal bot and an Elo rating of 1268. Notably, `PokéChamp` using Llama 3.1:8b outperforms PokéLLMon powered by GPT-4o, demonstrating the effectiveness of our approach even with smaller language models. Figure 7 illustrates the pairwise win rates between methods (left) and the impact of team composition on performance (right). To isolate the effects of team composition and the terastallize mechanic, we conducted additional experiments using mirror matchups without terastallize. Table 3 presents these results. These results suggest that team composition significantly impacts performance, with `PokéChamp` achieving a 90% win rate against Abyssal in mirror matchups.

**Against human players on the online ladder.** To assess real-world performance, we evaluated `PokéChamp` on the Pokémon Showdown online ladder between September 1st and September 7th, 2024, where it faced human opponents in the Gen 9 OU format. Each player has 2 minutes and 30 seconds to play the game and the player will lose immediately if one fails to make a decision within the time limit (similar to a chess clock). Players can receive up to 15 additional seconds added to their time per move. For about *one third* of the games, `PokéChamp` lost by exceeding the turn time limit. Within the remaining two thirds of games, `PokéChamp` achieved a 76% win rate. After 50 games, `PokéChamp` reaches an Elo score above 1300 on the online Showdown ladder. If removing losses due to timeout, we can estimate its Elo rating, where `PokéChamp` will achieve an Elo rating of 1300-1500, which places it in the top 30%-10% of players. We then use the win rates of each bot (PokéLLMon, Abyssal, etc.) against `PokéChamp` to estimate the Elo range of each method in Figure 1.

On March 3rd, 2025, we investigated a speed-optimized variant of our method, `PokéChamp-Fast`, where the LLM dynamically chose between using the calculated one-step lookahead action or performing a truncated minimax search based on the current game state. This was designed to alleviate time constraint issues. However, the resulting Elo

performance ranged between 1150-1310 without convergence, a decrease compared to our standard method. We attribute this performance difference to a covariate shift in the competitive metagame over time. Specifically, the LLM's pre-training data, which has a fixed cutoff date, likely biases its decision-making even when provided with updated historical statistics about the current metagame. This suggests that the LLM's inherent prior knowledge can outweigh real-time statistical information, particularly when deciding between strategic exploration (minimax search) and exploitation (one-step lookahead).

We observed that `PokéChamp` struggled against two specific strategies: stall tactics and excessive switching. These challenges likely stem from the limited lookahead depth necessitated by time constraints and the difficulty of accurately modeling such strategies in-context. Further analysis of these weaknesses is provided in Appendix C.1 and C.2. During live demos, we note that games played against humans who knew they were playing against `PokéChamp` were able to determine the excessive switching limitation and tailor an adversarial strategy to take advantage of these limitations. Thus, we emphasize anonymity for accurate performance analysis.

In summary, our evaluation demonstrates that `PokéChamp` achieves state-of-the-art performance in Pokémon battles, outperforming both heuristic and LLM-based baselines while exhibiting expert-level play against human opponents. These results underscore the effectiveness of our approach in leveraging LLMs for complex, partially observable game environments.

## 6. Related Work

**Competitive Games.** RL self-play has produced superhuman agents in Chess (Campbell et al., 2002; Silver et al., 2017), Go (Silver et al., 2016), Poker (Brown & Sandholm, 2019; 2018), StarCraft II (Vinyals et al., 2019), and Dota 2 (Berner et al., 2019), and has recently scaled to domains such as Street Fighter (Li et al., 2024) and Gran Turismo Sport (Wurman et al., 2022). For *Pokémon*, prior work combines self-play with heuristics to reach competitive strength (Huang & Lee, 2019; Whidden, 2023). In contrast, `PokéChamp` attains expert-level play without any explicit training.

**Language Agents for Games.** LLMs remain weak planners—for example, they still miss the Nash strategy in Tic-Tac-Toe (Topsakal & Harper, 2024)—but a growing literature explores augmentations (Hu et al., 2024a). Prompt-based agents have been demonstrated for *Pokémon* (Hu et al., 2024b), StarCraft (Ma et al., 2023), and Avalon (Shi et al., 2023; Stepputtis et al., 2023); tool-use extensions tackle NetHack (Küttler et al., 2020; Jeurissen et al., 2024); open-

world settings include Minecraft (Wang et al., 2023) and Spring (Wu et al., 2024); and finetuning on human transcripts enables Diplomacy play (, FAIR). Complementary approaches study distillation (Nalty & Rosenthal, 2024), online self-play finetuning (Zhou et al., 2024), and reward shaping with LLM feedback (Klissarov et al., 2023).

**Prompting and Planning.** Prompt strategies—chain-of-thought (Wei et al., 2022), self-consistency (Wang et al., 2022), Tree-of-Thoughts (Yao et al., 2024), and ReAct (Yao et al., 2022)—improve reasoning, while search and RL further enhance control. RAP treats an LLM as both planner and world model (Hao et al., 2023); TS-LLM applies AlphaZero-style tree search during decoding (Feng et al., 2023); and commonsense planners query LLMs as knowledge sources (Zhao et al., 2023). Self-correction methods such as SCoRe (Kumar et al., 2024) and Reflexion (Shinn et al., 2024) refine policies via linguistic feedback without heavy finetuning.

## 7. Conclusion

In this paper, we introduce `PokéChamp`, which augments minimax tree search with the following LLM-based components: (1) action sampling, (2) opponent modeling, and (3) a state value function. `PokéChamp` achieves state-of-the-art performance against heuristic and LLM-based bots and expert performance against real players on the online ladder. The objective of our framework lies at the intersection of imitation learning, best response estimation, and Nash equilibrium approximation. While imitation learning estimates the best response to the meta-game, and computing the exact Nash policy may be intractable, `PokéChamp` aims to strike a balance between these approaches. Our max-min formulation in the minimax search finds a conservative action that approximates the true best response, although the exact relationship between our method and the optimal best response remains an open question for future investigation. Further performance enhancements are currently limited by the accuracy of opponent modeling and the method's online computational budget. By increasing the breadth and depth size of the search, we expect to see further improvement's to the performance of the method. Additionally, our work can be taken advantage of adversarially due to static opponent modeling. Our work leaves open challenges in opponent modeling and generative minimax planning for future work exploring competitive multi-agent settings and *future superhuman performance* in Pokémon battling. Our benchmark explores LLMs and test time planning for competitive POMG in Pokémon battling. However, we provide a generalized framework of action sampling, one step lookahead world modeling, opponent modeling, and value that may be easily applied to other frameworks.

## Acknowledgement

The authors acknowledge the support of Office of Naval Research Grant N00014-22-1-2253, National Science Foundation Grant NSF-OAC-2411299, the National Science Foundation Graduate Research Fellowship Program under Grant No. DGE-2039656, and computational resources from Princeton Language and Intelligence (PLI).

## Impact Statement

In our work, we use a small number of games with our method on the community-maintained open-source competitive Pokémon platform, Pokémon Showdown. As the popularity of AI agents for Pokémon increases, we urge researchers to limit use of the online ladder evaluation to avoid data contamination and overwhelming human users of the platform with AI opponents. Rather, we recommend that researchers compete AI agents against other AI agents.

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

# A. Appendix

# B. Background

### B.1. Pokémon Showdown and Competitive Battling

Pokémon Showdown is an online battle simulator that allows players to engage in competitive Pokémon battles without the need for in-game breeding or training. It implements the official battle mechanics and provides a platform for various competitive formats.

In competitive Pokémon battles, two players each bring a team of up to six Pokémon and engage in turn-based combat. The objective is to make all of the opponent's Pokémon faint. Each turn, players can choose to use one of their active Pokémon's moves, switch to a different Pokémon, or use an item.

Key aspects of competitive battling include:

- **Type effectiveness:** Each Pokémon and move has one or two types, with a complex system of type matchups affecting damage calculations.

- **Stats:** Six core stats (HP, Attack, Defense, Special Attack, Special Defense, and Speed) determine a Pokémon's performance in battle.

- **Abilities:** Special traits that can provide various effects during battle.

- **Held items:** Objects that Pokémon can carry to gain additional effects or bonuses.

### B.2. Battle Formats

Pokémon Showdown supports various battle formats, including:

- **Singles:** One-on-one battles where each player has one active Pokémon at a time.

- **Doubles:** Battles where each player has two active Pokémon simultaneously.

- **Random Battles:** Players are assigned random teams of fully-trained Pokémon.

- **Tiered formats:** Formats that group Pokémon based on their usage and perceived strength (e.g., OU, UU, RU).

Our research focuses primarily on the Generation 9 OverUsed (OU) format, which is a singles format featuring the most commonly used Pokémon that are not considered too powerful for standard play.

### B.3. Partial Observability in Pokémon Battles

Pokémon battles are partially observable Markov games. The partial observability stems from several factors:

- Players can only see their opponent's active Pokémon, not their entire team.

- The exact stats, abilities, and movesets of the opponent's Pokémon are unknown until revealed through battle actions.

- Some moves and abilities can temporarily change the game state in ways that are not fully observable to the opponent.

This partial observability adds a layer of complexity to decision-making and strategy, as players must make inferences about their opponent's team and potential actions based on limited information.

### B.4. Pokémon Battling AI

Various approaches have been developed for creating AI agents capable of playing Pokémon battles:

- **Rule-based systems:** These include the AI used in official Pokémon games, such as the Abyssal bot, which follows pre-defined heuristics for decision-making.

- **Search-based methods:** Approaches like Expectiminimax have been attempted but face challenges due to the large branching factor and enforced time limits per turn.

- **Machine learning approaches:** Some efforts have used supervised learning on battle data to predict opponent moves and inform decision-making.

- **Reinforcement learning:** While successful in other game domains, the complexity and partial observability of Pokémon battles have made this approach challenging.

Recent work has explored the use of large language models (LLMs) for Pokémon battling, such as PokéLLMon (Hu et al., 2024b), which uses prompting techniques to generate actions based on battle state descriptions.

### B.5. Evaluation Metrics

The primary metrics for evaluating Pokémon battling agents include:

- **Win rate:** The percentage of games won against a specific opponent or set of opponents.

- **Elo rating:** A relative skill rating system used in competitive gaming, where a higher Elo indicates stronger performance against other rated players.

- **Average number of turns:** The typical length of games played by an agent, which can indicate efficiency or playstyle.

These metrics allow for comparison between different AI agents and human players, providing insights into the relative strengths and weaknesses of various approaches.

### B.6. Damage Calculation

The damage calculator is a crucial component of `PokéChamp` which helps use the LLM predicted information to get the one step lookahead, implementing a key part of the transition function $P_h(s_{h+1}|s_h, a_h, b_h)$. It uses the following equation to compute the expected damage for a given move:

$$
\begin{aligned}
\text{Damage} = &\left( \frac{1}{50} \left( \left( \frac{2}{5} \cdot \text{Level} + 2 \right) \cdot \text{Power} \cdot \frac{A}{D} \right) + 2 \right) \\
&\cdot \text{Targets} \\
&\cdot \text{PB} \\
&\cdot \text{Weather} \\
&\cdot \text{GlaiveRush} \\
&\cdot \text{Critical} \\
&\cdot \text{random } (0.85, 1.00] \\
&\cdot \text{STAB} \\
&\cdot \text{Type} \\
&\cdot \text{Burn} \\
&\cdot \text{other} \\
&\cdot \text{ZMove} \\
&\cdot \text{TeraShield.}
\end{aligned}
\tag{2}
$$

where:

- $A$ and $D$ are the relevant attack and defense stats

- Targets is 0.75 if the move hits multiple targets in doubles/triples, 1 otherwise

- Weather is 1.5 for Water moves in Rain or Fire moves in Sun, 0.5 for the opposite, 1 otherwise

- Critical is 1.5 for critical hits, 1 otherwise

- STAB (Same Type Attack Bonus) is 1.5 if the move type matches the user's type, 1 otherwise

- Type is the type effectiveness multiplier (0.25, 0.5, 1, 2, or 4)

- Burn is 0.5 for physical moves if the user is burned, 1 otherwise

- Other includes various move-specific and ability-based modifiers

This detailed calculation allows `PokéChamp` to accurately estimate the outcomes of different actions, informing its decision-making process within the minimax tree search.

An example prompt that is generated by the damage calculator for all matchups is provided in listing 1.

```
1  Requires switch:
2  dragapult vs. primarina:
3  dragapult outspeeds primarina
4  dragapult's moves:
5  dragondarts: 161 turns to KO opponent's pokemon
6  uturn: 6 turns to KO opponent's pokemon
7  quickattack: 5 turns to KO opponent's pokemon
8  terablast: 5 turns to KO opponent's pokemon
9  dragapult's moves if opponent's primarina uses 'terastallize':
10 dragondarts: 3 turns to KO opponent's pokemon
11 uturn: 11 turns to KO opponent's pokemon
12 quickattack: 321 turns to KO opponent's pokemon
13 terablast: 321 turns to KO opponent's pokemon
14 dragapult's moves if it uses 'terastallize' and opponent's primarina uses '
       terastallize':
15 dragondarts: 4 turns to KO opponent's pokemon
16 uturn: 6 turns to KO opponent's pokemon
17 quickattack: 10 turns to KO opponent's pokemon
18 terablast: 9 turns to KO opponent's pokemon
19 dragapult's moves if it uses 'terastallize' and opponent's primarina does NOT use '
       terastallize':
20 dragondarts: 161 turns to KO opponent's pokemon
21 uturn: 6 turns to KO opponent's pokemon
22 quickattack: 5 turns to KO opponent's pokemon
23 terablast: 5 turns to KO opponent's pokemon
24 Opponent moves: primarina
25 moonblast: 2 turns to KO your pokemon
26 psychicnoise: 4 turns to KO your pokemon
27 surf: 4 turns to KO your pokemon
28 flipturn: 10 turns to KO your pokemon
```

*Listing 1.* Example prompt from damage calculator.

## C. More Puzzles

To further evaluate `PokéChamp`'s strategic capabilities and identify areas for improvement, we developed additional puzzle scenarios that test specific aspects of Pokémon battling.

## C.1. Stall Strategy

Stall strategies in Pokémon battles involve using defensive Pokémon with high HP and recovery moves to gradually wear down the opponent. These strategies often rely on status effects, entry hazards, and passive damage to win battles over many turns.

`PokéChamp` struggles with stall strategies due to the uncertainty they introduce in the current matchup. This uncertainty often causes `PokéChamp` to switch its current Pokémon frequently, which can be counterproductive. The agent's difficulty in handling stall strategies stems from two main factors:

1. Limited lookahead: The minimax tree search used by `PokéChamp` may not extend far enough to fully capture the long-term benefits of maintaining position against a stall team.

2. Overemphasis on immediate damage: The value function $V_K(s_K)$ may not adequately account for the cumulative effects of status conditions and passive damage over many turns.

Figure 8 (left) illustrates this behavior. In this scenario, `PokéChamp` initially chooses Darkrai against Blissey, recognizing that Darkrai's Focus Blast is strong against Blissey. However, the agent then decides to switch to Enamorus, which faints from entry hazards. After sending Darkrai back in, it misses the first Focus Blast against Blissey. This miss increases uncertainty in `PokéChamp`'s decision-making process, leading it to switch to another Pokémon rather than maintaining its position.

## C.2. Excessive Switching

Another challenging scenario for `PokéChamp` is when opponents employ excessive switching strategies. In this approach, opponents frequently switch their Pokémon to disrupt the agent's planning and exploit the limitations of short lookahead methods.

`PokéChamp` struggles to capitalize on or defend against this strategy due to several factors:

1. Incomplete opponent modeling: The opponent policy $\nu_h(\cdot|y_h)$ may not accurately capture the likelihood of frequent switching.

2. Myopic decision-making: The limited depth of the minimax tree search may prevent `PokéChamp` from recognizing the long-term advantages of predicting and punishing switches.

3. Overcommitment to predicted optimal moves: Once `PokéChamp` identifies a strong move against the current opponent Pokémon, it may persist in using that move even when it becomes suboptimal due to switches.

Figure 8 (right) demonstrates this weakness. In this example, `PokéChamp` consistently chooses Focus Blast, a Fighting-type move, even as the opponent switches between two Pokémon that resist Fighting-type attacks. This behavior allows the opponent to exploit `PokéChamp`'s predictability and inability to adapt to the switching strategy.

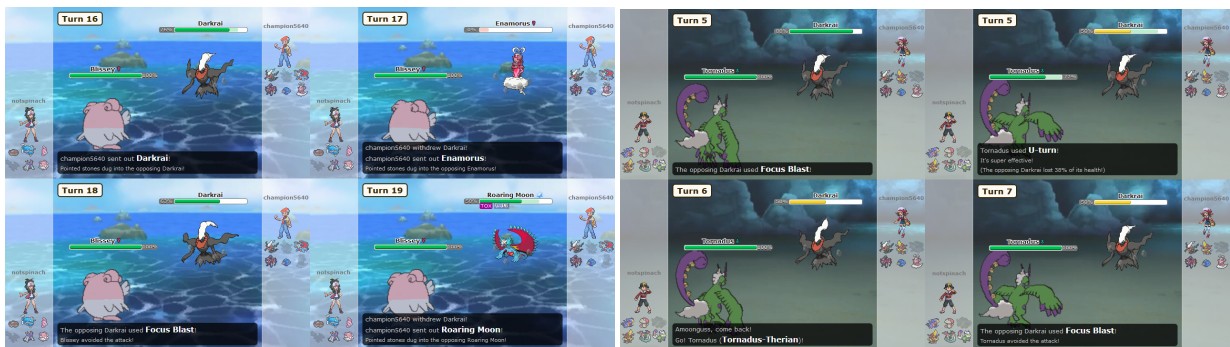

Figure 8. **Left:** *Stall strategy*: `PokéChamp`'s response to a stall tactic, illustrating its difficulty in maintaining a consistent strategy against defensive play. **Right:** *Opponent excessive switching strategy*: `PokéChamp`'s struggle to adapt to an opponent's frequent Pokémon switches, demonstrating its tendency to persist with suboptimal move choices.

These puzzles highlight areas where `PokéChamp`'s decision-making process can be improved, particularly in handling

long-term strategies and adapting to dynamic opponent behaviors.

## D. Random Battles

To evaluate `PokéChamp`'s performance in a setting with higher uncertainty and variability, we conducted experiments in the Gen 8 Random Battles format. This format presents unique challenges as players must adapt to randomly assigned teams, testing the agent's ability to quickly assess team strengths and weaknesses.

We compared `PokéChamp` against various baselines in two scenarios: with and without the dynamax mechanic. The dynamax mechanic allows Pokémon to temporarily increase their power and HP, adding another layer of strategic complexity to battles.

### D.1. Results without Dynamax

Table 4 presents the results for Gen 8 Random Battles without the dynamax mechanic.

*Table 4.* Gen 8 Random Battles without dynamax mechanic.

| Bot Method | Language Model | Win Rate vs. Abyssal (%) |
|---|---|---|
| **PokéChamp** | **GPT-4o** | **70** |
| PokéChamp | Llama 3.1:8b | 64 |
| PokéLLMon (Hu et al., 2024b) | GPT-4o | 56 |
| One Step Lookahead | N/A | 44 |

In this setting, `PokéChamp` with GPT-4o achieves a 70% win rate against the Abyssal bot, significantly outperforming other methods. Notably, `PokéChamp` using the smaller Llama 3.1:8b model still outperforms PokéLLMon with GPT-4o, demonstrating the effectiveness of our approach even with less powerful language models.

### D.2. Results with Dynamax

Table 5 shows the results for Gen 8 Random Battles with the dynamax mechanic enabled.

*Table 5.* Gen 8 Random Battles with dynamax mechanic.

| Method | LLM | Win Rate vs. Abyssal (%) | Elo | Avg. # Turns |
|---|---|---|---|---|
| **PokéChamp** | **GPT-4o** | **56** | **1273** | **17.1** |
| PokéChamp | Llama 3.1:8b | 52 | 1184 | 19.1 |
| PokéLLMon | GPT-4o | 36 | 1048 | 22.5 |
| Abyssal | N/A | N/A | 1213 | 19.0 |
| One Step Lookahead | N/A | 16 | 998 | 18.9 |
| Max Power | N/A | 4 | 787 | 23.2 |
| Random | N/A | 0 | 493 | 24.3 |

With dynamax enabled, `PokéChamp` maintains its superior performance, achieving the highest Elo rating of 1273 and the shortest average game length of 17.1 turns. This indicates that `PokéChamp` can effectively utilize the dynamax mechanic to gain advantages and close out games more quickly.

### D.3. Analysis of Results

Figure 9 provides a detailed view of the pairwise matchups between different methods in Gen 8 Random Battles.

These results demonstrate several key points:

1. **Adaptability**: `PokéChamp`'s strong performance in Random Battles highlights its ability to quickly assess and adapt to

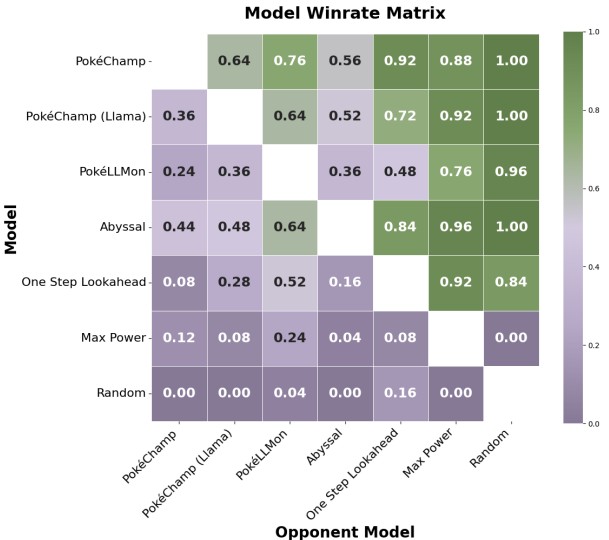

*Figure 9.* Gen 8 Random Battles matchup matrix per method.

unfamiliar team compositions, a crucial skill in this format.

2. **Effective use of dynamax**: The maintained performance with dynamax enabled suggests that `PokéChamp` can effectively incorporate this mechanic into its strategy, likely due to the LLM's understanding of its impact on the game state.

3. **Efficiency**: The shorter average game length for `PokéChamp` indicates that it can identify and execute winning strategies more quickly than other methods.

4. **Robustness to randomness**: The consistent outperformance of other methods, even with the added randomness of team composition, demonstrates the robustness of `PokéChamp`'s decision-making process.

5. **LLM size trade-off**: While GPT-4o provides the best performance, the strong results with Llama 3.1:8b suggest a favorable trade-off between model size and performance for resource-constrained applications.

These experiments in the Random Battles format further validate the versatility and effectiveness of `PokéChamp`, showing that its performance advantages extend beyond fixed team compositions to scenarios with higher uncertainty and variability.

