# OpenReview forum: "PokéChamp: an Expert-level Minimax Language Agent"
_ICML.cc/2025/Conference — ICML 2025 spotlightposter_

### Official Review · Reviewer_UL1n · 2025-03-11

**Overall Recommendation:** 3

**Summary:**

This paper introduces PokeChamp, and LLM combined with a game-playing agent to perform minimax search for winning Pokemon battles. The authors replace several parts of minimax search with an LLM and introduce a Pokemon battling dataset to understand LLM agent's failures. PokeChamp is able to reach the top 90% of human performnace.

**Claims And Evidence:**

See strengths and weaknesses

**Essential References Not Discussed:**

See strengths and weaknesses

**Experimental Designs Or Analyses:**

See strengths and weaknesses

**Methods And Evaluation Criteria:**

See strengths and weaknesses

**Other Comments Or Suggestions:**

Nit: change GPT 4-o to 4o everywhere (I noticed this in the abstract) and change Llama3.1:8b to Llama-3.1-8B (L320). Also are you using the instruct version? If so that should be mentioned.

**Other Strengths And Weaknesses:**

# Strengths
- The paper provides an interesting exploration of how to use LLM agents for partially observable games. There are several ideas that were explored that could point towards future paths to understanding how to mix LLMs with more classical AI systems.
- It is clear that the authors enjoyed creating their agent, which makes the paper fun to read.
- The authors collected a dataset which will greatly help enable progress towards studying partially observable games. Moreover, the authors use the dataset to quantify their agent's performance.

# Weaknesses
- **The LLM does not appear to be well utilized:**
    - The LLM appears in the one-step lookahead when estimating the opponent's stats, the opponent's action prediction, and the value function. In the first case, it appears that the estimating the opponent's stats from data likely can be learned from data and does not require a language prior. In the second case, the authors show that the LLM's performance is slightly above random (although this could possibly be due to Pokemon data being OOD for the LLM), and in the third case there is no evidence that the LLM's estimation of the value function is actually accurate (although I realize that this is difficult to evaluate).
    - While I think this work is still interesting as an exploration of how to combine LLMs with agents, I don't think the Pokemon setting is particularly illustrative for showcasing the strengths of the LLM. Much of this work could be done with standard DNNs.
- No ablation studies of the agent: would it be possible for the authors to ablate some of the LLM components of the agents to see where the performance gains are coming from? Some suggestions: a) replace the LLM in the stat estimation with a trained NN, b) remove the LLM from the action prediction (randomly predict an action or train a NN to predict the action), or c) use smaller LLMs, e.g., Llama-3.2-3B. How much does this tank performance?
- No studies of the agent's value function: although I realize that it's extremely difficult to even empirically estimate the value function, could the authors speak to the quality of the estimated value function produced by th eLLM?

**Questions For Authors:**

Is 90% ELO good in an absolute sense? I wonder if there's just not many people playing Pokemon battles or the average skill level of human players is pretty low. I'm guessing it would take someone much more time to get 90% at chess than at Pokemon battles. Do you have a sense of the difficulty of the accomplishment? (I still think it's a cool result regardless but it's a bit abstract for me).

**Relation To Broader Scientific Literature:**

See strengths and weaknesses

**Theoretical Claims:**

See strengths and weaknesses

---

> ### Author Rebuttal · Authors · 2025-03-31
>
> Thank you for recognizing the value of our exploration of LLM agents for partially observable games and appreciating our dataset contribution. These are indeed core strengths of our work.
>
> **LLM utilization**: LLMs are key components in our system to achieve the claimed performance. Our work represents an important attempt to effectively utilizing LLMs in complex two-player game environments: While opponent stat estimation could potentially be learned from data alone, LLMs provide crucial domain knowledge that would otherwise require extensive labeled data and model training. For instance, they understand type matchups, move effectiveness, and Pokemon meta-strategies with minimal prompting (as shown in Figure 5, page 6, where PokéChamp adapts to complex mechanics like Terastallization and Dynamax). The one-step lookahead mechanism combines LLM knowledge with damage calculations to determine the best action for the game by extrapolating over short horizon information.
>
> While opponent action prediction is challenging (13-16% accuracy for min-player, Table 1, page 5), this is still valuable for narrowing the search space in minimax and substantially better than random (which would be <1% given the large action space). Even imperfect opponent modeling significantly improves overall performance, as demonstrated by our 76% win rate against the strongest LLM-based bot and 84% against the strongest heuristic bot (Table 3, page 7).
>
> **Pokemon as illustrative settings for LLM**: Pokémon battle is a text-based two-player game featuring a vast pool of Pokémon, types, items, moves, and game mechanics, all described in text, resulting in a combinatorial explosion of possible configurations. It exemplifies a complex, partially observable environment with a massive state space (~10^354 for the first turn alone, page 2). This complexity makes it an excellent testbed for demonstrating how LLMs can constrain search to human-like strategies without explicit training, which is our core contribution.
>
> **ablation studies**: We appreciate this suggestion. We do provide a partial ablation by comparing PokéChamp with GPT-4o versus Llama 3.1:8b (Tables 3-4, page 7-8), demonstrating that our approach works well even with smaller models. Though frontier models such as GPT-4o perform best. We also compare against the One Step Lookahead agent without the full minimax framework (Tables 3-4), showing the value of our complete approach.
>
> Regarding the value function quality, as mentioned in our initial response, the best evaluation is performance in games. Our expert-level performance (1500 Elo rating, top 10%, Figure 1, page 1) demonstrates the effectiveness of our value function approximation. The value function is able to correctly prioritize actions that lead to winning strategies, as shown by our significantly higher win rates compared to all baselines.
>
> **the significance of 90% Elo**: Pokémon Showdown is a very active competitive ladder with roughly 3000 active games at any given time, and over millions of games occurring every month. This makes the top 10% achievement substantial. Reaching this level requires deep understanding of complex strategies, team compositions, and meta-game knowledge. The 1500 Elo rating on Pokémon Showdown is considered expert-level play among the community.
>
> We thank you for your suggestions regarding model naming conventions and will address these in our revision.

---

### Official Review · Reviewer_734n · 2025-03-13

**Overall Recommendation:** 3

**Summary:**

The paper introduces PokéChamp, an agent that leverages minimax-based search to play competitive Pokemon battles. Specifically, the LLM performs action sampling, opponent modeling and state value calculation, allowing it to navigate partially observable state spaces of the battles. The authors also present various experiments in the paper showcasing that PokéChamp outperforms existing LLM-base and heuristic bots built for Pokemon battles.

**Claims And Evidence:**

The claims in the paper are largely well supported. For instance, many experiments were conducted and empirical results showed that PokéChamp outperforms existing rule-based bots and LLM-based agents under various different conditions when measuring win rates, with the measured Elo ratings also reinforce these results.

However, the claim in the conclusion that this paper provides a generalized framework for other POMGs lack empirical evidence. The design of PokéChamp is highly specific to the mechanic of Pokemon battles, and hence while the framework is sound in theory, it is hard to tell how generalizable this method will be to other games.

**Essential References Not Discussed:**

N/A

**Experimental Designs Or Analyses:**

The battles that are conducted in the experiments span multiple setups and compare multiple baselines bots and human players to PokéChamp, providing a comprehensive evaluation. The metrics such as win rate, Elo and average number of turns are suitable measures of the performance of the framework.

The analysis of the weak points of PokéChamp (such as losses due to time, or handling stall tactics and excessive switching) were insightful in identifying what the limitations are and where the framework could be improved.

**Methods And Evaluation Criteria:**

The methods used do make sense for the application at hand. By integrating the three main minimax search steps: action sampling, opponent modeling and state value calculations, the agent can optimize decision making in the complex, partially observable environment. The one-step look ahead also helps improve decision making the estimating the immediate consequences of the potential actions, enabling PokéChamp to further improve the decisions made.

The evaluation that include comprehensive benchmarks against well known Pokemon bots and human players under diverse settings (which sometimes isolate particular game mechanics to test the agent’s understanding of those in particular) is robust. The metrics measured such as win rate and Elo are strong quantitative measures of the performance.

**Other Comments Or Suggestions:**

N/A

**Other Strengths And Weaknesses:**

Strengths:
- The paper presents a novel framework to integrate minimax search algorithms with LLM agents, enabling improved performance without the massive training requirements
- PokéChamp outperforms existing bots and agents across many benchmarks and battle configurations, proving its efficacy

Weaknesses:
- The biggest weakness of this paper is the limited generalization it has to other applications. Many components of PokéChamp are highly specific to Pokemon limits the ability for this framework to be easily applicable to other games without significant additional configurations (unlike more general RL based methods that can be generalized with less effort)
- Opponent modeling is mentioned as a key component of the framework, however the prediction accuracies are very low (13-16%) and hence more analysis into this step would be helpful to investigate the effectiveness of it

**Questions For Authors:**

- Can the authors speak more about the trade-offs associated with search depth and what their approaches are to optimize this balance?
- Can the authors speak more generally about the generalization of these methods beyond Pokemon battles?

**Relation To Broader Scientific Literature:**

The paper builds on the idea of developing AI that achieves superhuman performance in games, it draws reinforcement learning approaches from work such as AlphaZero, and extend those ideas by integrating LLMs into the framework to reduces the expensive training required in the existing frameworks. It also improves the research on other LLM based agents such as PokéLLMon by incorporating these planning algorithms. The combination of the two methodologies introduces ideas that can be used in further research, and are especially practical due to the lack of extensive training requirements.

**Theoretical Claims:**

The paper formulates Pokemon battles as partially observable Markov game (POMGs), but there are no explicit theoretical claims of proofs. In general, there is a slight gap between the theoretical game theory framework and the LLM generated value functions used in PokéChamp, as it is unclear how certain theoretical properties translate when using LLM-based approximations of the values.

---

> ### Author Rebuttal · Authors · 2025-03-31
>
> Thank you for highlighting our framework's novelty and effectiveness in integrating minimax search with LLMs, and for recognizing PokéChamp's strong performance across multiple benchmarks.
>
> **generalizability beyond Pokémon battles**: Our framework naturally applies to any two-player zero-sum competitive games beyond Pokemon where minimax tree search is feasible. As described in Section 3 (page 3-4), our framework implements three LLM-powered components that can generalize to any POMG: "(1) Action sampling via LLM and tool-assisted action generation, (2) Opponent modeling through historical data and LLM-based prediction, and (3) LLM-generated value function for leaf nodes. This method implements basic components of the minimax search tree with LLM predictions that can generalize to any POMG."
>
> The core innovation here is the replacement of traditional minimax components with LLM-based alternatives. While we demonstrate this in Pokémon, the structure is applicable to any two-player partially observable game where states can be described in natural language. Unlike RL-based methods that require extensive task-specific training, our approach only requires observation space descriptions in natural language. This makes implementation no more difficult than developing custom RL environments and reward functions.
>
> **search depth trade-offs**: As detailed in Section 5.3 (page 7), we implement strict time management to comply with the 15-second per turn limit: "Our search depth is limited by the 15 second time cutoff, which is a maximum of a two step lookahead with a branching factor of 16 (4 max player actions, 4 min player actions)." We address this trade-off dynamically throughout gameplay, as "the decision time when there is only 1 pokemon left is very fast due to the limited number of states to expand." This demonstrates our system's adaptability to different computational constraints.
>
> **opponent modeling accuracy**: While the raw prediction accuracies (13-16%) may appear low, this reflects the inherent challenge of predicting exact moves in a complex, partially observable environment with a large action space. Importantly, even imperfect opponent models provide significant value in minimax search by narrowing the branch exploration to more likely opponent responses. The comparative performances against state-of-the-art bots (76% win rate against the strongest LLM-based bot and 84% against the most advanced heuristic bot) demonstrate that our opponent modeling is effective despite these challenges.
>
> We appreciate your overall positive assessment of our work and hope our clarifications address your concerns about generalizability and search depth optimization.

---

### Official Review · Reviewer_yTKx · 2025-03-13

**Overall Recommendation:** 3

**Summary:**

This paper introduces PokeChamp, an LLM-powered game-theoretic agent designed for competitive Pokémon battles. PokeChamp uses LLM-guided minimax search to model decision-making in partially observable environments. It outperforms all prior LLM-based and heuristic-based Pokemon bots.

## update after rebuttal
The authors address most of my concerns, and I keep my original rating to lean toward accepting the paper.

**Claims And Evidence:**

1. The overhead of LLM-based search is not fully discussed, particularly its impact on real-time play.

2. While PokeChamp achieves the 90th percentile, there is a concern that a small fraction of players may be truly active. A comparison specifically with active players would provide a more realistic performance benchmark.

**Essential References Not Discussed:**

The idea of integrating LLMs with the minimax search framework for game-playing agents is closely related to prior work by Guo et al. (2024), which explores a similar concept in two-player zero-sum games. Guo, Wei, et al. "Minimax Tree of Thoughts: Playing Two-Player Zero-Sum Sequential Games with Large Language Models." ICML 2024 Workshop on LLMs and Cognition.

**Experimental Designs Or Analyses:**

Sensitivity to hyperparameters (e.g., search depth) is not explored.

Ablation studies on the effectiveness of each component are missing.

**Methods And Evaluation Criteria:**

The paper introduces Pokemon battle-specific terms (e.g., "Abyssal bot", "Elo") without detailed explanation or references. This makes the work less accessible to researchers unfamiliar with Pokémon battles.

**Other Comments Or Suggestions:**

See weaknesses and questions.

**Other Strengths And Weaknesses:**

**Additional Weaknesses:**

Prediction accuracy at higher Elo ratings: In Table 1, action prediction accuracy is highest at the 1800 Elo stage. Can the authors explain this phenomenon? Are skilled players' actions easier to predict?

Damage calculator inconsistency: In Appendix line 685, "dragondarts: 161 turns to KO Primarina" seems incorrect, as Primarina is Fairy-type and should be immune to Dragon-type moves. How does the damage calculator work?

**Questions For Authors:**

1. Can you go deeper into the online evaluation score, such as Elo? While PokéChamp achieves expert-level performance (90th percentile Elo), how does it compare to active players?

2. Can you provide the trend of decision time as the round progresses? At which step does the LLM’s response time exceed 15 seconds?

3. PokeChamp implements three LLM-based components (action sampling, opponent modeling, value estimation). Are these components specifically designed for Pokemon battles, or do they or their intuition have broader applicability?

**Relation To Broader Scientific Literature:**

Missing some discussion about related papers. see "Essential References Not Discussed".

**Theoretical Claims:**

Not much theory is included in this paper.

---

> ### Author Rebuttal · Authors · 2025-03-31
>
> Thank you for your thoughtful review. We appreciate your recognition of PokéChamp's strong performance against prior bots and human players.
>
> **Elo ratings and active players**: The Elo system on Pokémon Showdown (which we use for evaluation) only includes active players by design. Inactive accounts are reset to a base Elo of 1000 regularly and are not included in the percentile, ensuring our 90th percentile achievement reflects performance against the current active competitive community. As detailed in Figure 1 (page 1), PokéChamp's 1500 Elo rating places it firmly among expert players in this active ecosystem. This is comparable to how chess platforms like chess.com evaluate player performance. Pokemon Showdown has a smaller scale: roughly 3000 active games at any given time with millions of games played every month.
>
> **LLM overhead in real-time play**: We address this important constraint in Section 5.3 (page 7), noting that "33% of games were lost due to time constraints." Our system implements strict time management to ensure compliance with the 15-second per turn limit, which restricts our search depth to a maximum of two-step lookahead with a branching factor of 16 (4 max player actions, 4 min player actions). In practice, the decision time varies based on game state complexity significantly faster when fewer Pokémon remain in play due to the reduced state space. We have further developed a faster version of this using better coding practices and systems techniques to increase speed that we will release with our code that achieves the same performance as mentioned in the paper.
>
> **Pokémon-specific terms**: We agree that better explanation of Pokémon-specific terms would improve accessibility. Elo is a standard rating system widely used in competitive gaming beyond Pokémon (originated in chess), and we define the Abyssal bot in Section 5.1 (page 6) as "a rule-based heuristic bot used in official Pokémon games." We'll expand these definitions in the revised version.
>
> **Higher prediction accuracy at 1800+ Elo**: This interesting observation likely stems from the narrower strategy space employed by elite players. At this high level, players tend to favor optimal, established strategies rather than unpredictable or sub-optimal plays, making their decisions more consistent and thus more predictable.
>
> **Damage calculator**: The damage calculator example you noted (dragondarts: 161 turns) is actually a technical implementation detail. We cap the maximum turns to KO for cases where a Pokémon is effectively immune to an attack (as with Dragon-type moves against Fairy-types). By avoiding infinite values, we find that the LLM performs better at understanding and comparing all options.
>
> **Three LLM-based components**: Our three LLM-based components (action sampling, opponent modeling, and value estimation) form a general framework that is naturally applicable to any two-player partially observable game, not just Pokémon. These components could be adapted to other strategic games with similar characteristics.
>
> **Remaining comments**: We will address the missing ablation studies and hyperparameter sensitivity analysis description in our revised manuscript. We provide a link to a double-blind project website within our submitted project. The code link within the website is already announced “not for anonymous review” during the review period, but is available to reviewers after the review period is over. Thank you for your valuable feedback that will help us improve the final version.

---

> > ### Comment · Reviewer_yTKx · 2025-04-07
> >
> > Thanks to the authors for the detailed response to the concerns. I have no other concerns and think this work brings a contribution to the related community. So I keep the positive rating.

---

### Official Review · Reviewer_NRnt · 2025-03-20

**Overall Recommendation:** 4

**Summary:**

The authors introduce a novel RL agent that integrates and LLM into the tree-search process showing that their method can provide acceptable decisions in complex game states.

**Claims And Evidence:**

The authors claim that their method is SOTA on pokemon which they test in multiple ways and against multiple other models. This is a clear and strong evidence for the claim.

**Essential References Not Discussed:**

No

**Experimental Designs Or Analyses:**

See above, and I am also concerned with the validity of using an LLM as a black box. This significantly reduces the generalizabilty of the results as we do not know how much of the uplift came from the LLM's prompting, or some components of the LLM's training data (i.e. there are many games online that might have been included).

**Methods And Evaluation Criteria:**

Elo and winrate are not perfect metrics of RL search performance, but are the standard in these types of zero-sum games. So I don't have an issue with the evaluation. I would have liked some more detailed analysis of the model's behaviour (beyond section C), for example it might be exploiting defects in the other models, or taking very unusual lines against humans leading to an inflated winrate.

Can the model output a probability distribution over possible moves (policy map)? Looking at perplexity on a dataset of human moves, or the sharpness of the distribution would reveal more about the underlying reasoning.

**Other Comments Or Suggestions:**

See above

**Other Strengths And Weaknesses:**

See above

**Questions For Authors:**

See above

**Relation To Broader Scientific Literature:**

This is relevant to both RL and the larger project of integrating LLMs into task planning. As I said above the black box nature of the work is a sever limitation, but I think this helps highlight ways that LLMs can be used as a component of a larger planning system.

**Theoretical Claims:**

N/A, this is empirical work

---

> ### Author Rebuttal · Authors · 2025-03-31
>
> **More detailed model behavior analysis beyond Section C**: In addition to Section C, our paper provides the following analyses on our model/method with respect to the mechanics and strategies present in this game. In Section 4.3 (page 5-6), we present benchmark puzzles specifically designed to test PokéChamp's strategic decision-making abilities with special mechanics like Terastallization and Dynamax. Figure 5 illustrates how PokéChamp demonstrates understanding of complex type matchups and strategic mechanics usage rather than exploiting defects. Additionally, we evaluate against both bots and human players (Section 5.3), showing that our approach generalizes beyond potential weaknesses in other models.
>
> **probability distribution over possible moves**: We assess this in Table 1 (page 6), where we evaluate action prediction accuracy from Top-1 through Top-5 (ranking according to the action probabilities), effectively showing a distribution over moves compared to actual human actions.There may be multiple valid strategies at any point, which is why we include the Top-K metrics. The player prediction accuracy for PokéChamp varies between 26-30% for Top-1 and improves to 43-66% for Top-5 as Elo increases. We appreciate your suggestion to look at perplexity on human moves, which we can incorporate in future work.
>
> **the "black box" nature of using an LLM**: We have taken great care to make our approach transparent. As illustrated in Figure 2 (page 3), we explicitly replace three components of minimax search with LLM-based generations and clearly explain the contribution of each component. The LLM serves as a prior to constrain the search space to human-like strategies (page 2), leveraging general knowledge rather than Pokemon-specific training. Our ablation studies comparing PokéChamp with Llama 3.1 versus GPT-4o (Tables 3-4) demonstrate that some performance gains come from the intrinsic model capacity. However, we also provide ablations to show that all tested models greatly benefit from our methodology (comparing PokéChamp with PokéLLMon, Tables 3-4). In fact, being able to switch in the latest frontier model or open-source model when new capabilities emerge is an important benefit of this method.
>
> We also demonstrate robustness across different formats (Gen 8 Random Battles, Gen 9 OU) and against human players (achieving 1500 Elo, top 10% of players), showing that our approach generalizes well beyond specific test environments.
>
> Thank you for the positive feedback on our evaluation metrics and recognition of our work's relevance to both RL and LLM integration into task planning.

---

### Decision · Program_Chairs · 2025-05-01

**Decision:**

Accept (spotlight poster)

**Comment:**

The paper presents an innovative agent, PokéChamp, that integrates LLM-driven components into a minimax tree search to address the challenges of competitive Pokémon battles with partially observable states. The approach effectively combines action sampling, opponent modeling, and state value evaluation, leading to impressive performance improvements over traditional heuristic and LLM-based bots, as evidenced by robust win rate and Elo rating benchmarks. Although the empirical results within the Pokémon domain are convincing, the broader claim regarding the generalizability of the framework to other partially observable, two-player zero-sum games remains more speculative. Overall, the submission is a well-executed blend of classical AI and modern language model techniques, contributing a valuable dataset and novel insights.